# Development of an RNase H2 Activity Assay for Clinical Screening

**DOI:** 10.3390/jcm12041598

**Published:** 2023-02-17

**Authors:** Marian Simon Schulz, Cay Bennet Sartorius von Bach, Emilija Marinkovic, Claudia Günther, Rayk Behrendt, Axel Roers

**Affiliations:** 1University Hospital for Children and Adolescents, University of Leipzig, 04103 Leipzig, Germany; 2Institute for Immunology, University Hospital Heidelberg, 69120 Heidelberg, Germany; 3Department of Dermatology, University Hospital Carl Gustav Carus Dresden, TU Dresden, 01307 Dresden, Germany; 4Institute of Clinical Chemistry and Clinical Pharmacology, University Hospital Bonn, 53127 Bonn, Germany

**Keywords:** RNase H2, fluorometric assay, clinical screening, systemic sclerosis, systemic lupus erythemathosus, assay validation, interferonopathy, biomarker, DNA damage, chronic inflammation, cancer

## Abstract

As the key enzyme mediating ribonucleotide excision repair, RNase H2 is essential for the removal of single ribonucleotides from DNA in order to prevent genome damage. Loss of RNase H2 activity directly contributes to the pathogenesis of autoinflammatory and autoimmune diseases and might further play a role in ageing and neurodegeneration. Moreover, RNase H2 activity is a potential diagnostic and prognostic marker in several types of cancer. Until today, no method for quantification of RNase H2 activity has been validated for the clinical setting. Herein, validation and benchmarks of a FRET-based whole-cell lysate RNase H2 activity assay are presented, including standard conditions and procedures to calculate standardized RNase H2 activity. Spanning a wide working range, the assay is applicable to various human cell or tissue samples with overall methodological assay variability from 8.6% to 16%. Using our assay, we found RNase H2 activity was reduced in lymphocytes of two patients with systemic lupus erythematosus and one with systemic sclerosis carrying heterozygous mutations in one of the *RNASEH2* genes. Implementation of larger control groups will help to assess the diagnostic and prognostic value of clinical screening for RNase H2 activity in the future.

## 1. Introduction

An increasing number of inflammatory and degenerative diseases are found to be associated with compromised genome integrity. In some of these, genome damage is assumed to be a central pathogenic event, while in others DNA damage may represent an epiphenomenon [1,2,3,4,5]. Resolution of RNA/DNA hybrids is central to various DNA transactions and maintenance of genome integrity. Mammals express RNases H1 and H2, which both cleave RNA/DNA hybrids by catalyzing phosphodiester bond hydrolysis [6]. The enzymes play a role in resolution of R-loops, maturation of Okazaki fragments, repression of endogenous retroelements and degradation of RNA/DNA hybrids during cell death [7,8,9]. While RNase H1 requires hybrids with at least four consecutive ribonucleotides, RNase H2 also cleaves single ribonucleotides embedded in the DNA double helix [10,11]. Ribonucleotides are incorporated into genomic DNA in very high numbers during replication due to the limited capacity of the replicative polymerases to discriminate them from deoxyribonucleotides [12,13]. To prevent destabilization of DNA, ribonucleotides are rapidly removed post-replication by the ribonucleotide excision repair (RER) pathway initiated by RNase H2-mediated nicking 5′ of the ribonucleotide. RER is the only error-free pathway capable of removing single ribonucleotides from DNA [14]. Failure to repair these lesions leads to DNA damage [8,13,15,16,17,18,19]. In mammals, complete loss of RNase H2 activity leads to embryonic lethality [16,18]. Partial loss of function, however, caused by hypomorphic *RNASEH2* alleles can lead to autoinflammation and autoimmunity, as for example in Aicardi-Goutières syndrome (AGS), a monogenic ‘type I interferonopathy’ [20,21,22]. Hypomorphic *RNASEH2* alleles also contribute to the polygenic predisposition for systemic lupus erythematosis (SLE) [23,24,25]. Investigation of RNase H2-deficient human cells and mice recently led to elucidation of an important link between genome damage and chronic inflammation [16]. DNA lesions ensuing from unrepaired ribonucleotides result in chromosomal aberrations, problems of mitotic segregation of defective chromosomes and formation of micronuclei. Upon collapse of the unstable micronuclear envelope, micronuclear chromatin is sensed by the intracellular DNA sensor cGAS, in turn resulting in activation of the sensor STING and activation of type I IFN and proinflammatory cytokine responses [26,27,28]. Chronic activation of cGAS/STING signaling leads to autoinflammation, loss of T cell and B cell tolerance and autoimmune pathology [21]. RNase H2 deficiency also predisposes to cancer in mice [29,30] and RNase H2 loss-of-function mutations occur in large fractions of human chronic lymphocytic leukaemia and prostate cancer [31]. Reduced expression of RNase H2 is associated with reduced survival in colonic cancer [30]. Conversely, upregulation of RNase H2 subunits was found to be a malignancy factor in numerous carcinomas and sarcomas [32,33,34]. Moreover, double-strand breaks resulting from compromised RNase H2 function were reported to contribute to neurodegeneration and ageing [35,36,37]. Collectively, RNase H2 is a relevant diagnostic and prognostic factor in diverse human disease settings, warranting clinical testing for RNase H2 activity in human cells or tissues. 

Human RNase H2, unlike its monomeric prokaryotic isoenzyme RNase HII, is a heterotrimeric complex consisting of three proteins, the catalytic subunit RNase H2A and two auxiliary subunits, RNase H2B and RNase H2C [38,39,40]. About 50 disease-causing *RNASEH2* variants have been identified to date [24,25,27,41], most of which are located in subunit B. While many variants exhibit reduced RNase H2 substrate binding and hydrolysis, other mutant proteins did not show impaired activity in cell-free assays using recombinant enzymes [42]. The latter might feature compromised complex stability or interaction with additional proteins in vivo. 

Although measurement of RNase H activity in mammalian cell samples has been performed since its discovery in 1969 [43], a standardized and validated method available for clinical use has been lacking. RNase H activity can be quantified by several different approaches relying on acid-insoluble precipitation, gel electrophoresis or HPLC. Two groups developed a fluorescence assay suitable for high-throughput studies and superior to earlier approaches with respect to precision, speed, labor and cost. RNase H2-mediated cleavage of a double-stranded DNA substrate containing a single ribonucleotide results in release of a fluorescein-labelled fragment from a quencher [44]. Herein, we adapt this assay into a standardized and validated procedure relying on whole cell lysates for clinical screening of effective intracellular RNase H2 activity. 

## 2. Materials and Methods

### 2.1. Ethics Approval and Control Group Selection

Ethics approval was granted by the ethics committee of the Medical Faculty Carl Gustav Carus, TU Dresden (EK 31022012, EK 169052010). Volunteers older than 18 years of age without overt disease for the past two weeks were included after informed consent. Pregnancy or medication, abuse of alcohol or drugs were exclusion criteria. Volunteers did not receive financial or other compensation. 

### 2.2. Cell Culture

HeLa cells and murine embryonic fibroblasts (MEFs) were cultured in Gibco^®^ DMEM-Dulbecco’s Modified Eagle Medium (Fisher Scientific GmbH, Schwerte, Germany) at 37 °C and 5% CO_2_. For harvesting, the medium was aspirated and adherent cells were washed twice with 1× PBS followed by incubation with 1× trypsin (0.25%, Life Technologies Germany, Darmstadt, Germany) at 37 °C for 2 min. Digestion was stopped by addition of an FCS-containing medium; cells were detached by pipetting, transferred into a 15-mL conical tube and pelleted at 330× *g* for 5 min. Cells were resuspended and washed twice in 5 mL of chilled PBS for freezing, or in 1× FACS buffer for FACS sorting. For freezing, the supernatant was discarded, and pellets were shock-frozen in liquid nitrogen and then stored at −80 °C for a maximum of 4 weeks.

### 2.3. Isolation of Primary Cells from Human Blood and Mice 

For isolation of human peripheral blood mononuclear cells (PBMC), peripheral blood was collected in 10 mL heparinized tubes, stored at 4 °C and analyzed within 4 h. Blood was diluted in an equal volume of PBS (calcium- and magnesium-free, equilibrated to room temperature (RT)). PBMC were isolated by standard Ficoll^®^-Paque density gradient centrifugation and washed 3 times with chilled PBS. Murine keratinocytes, peritoneal cells, spleen cells and embryonic fibroblasts were isolated from C57BL/6 mice by standard procedures [45,46,47,48].

### 2.4. Short-Time Culture of Spleen B Cells from FUCCI Mice 

Total spleen cells of three *Rnaseh2b* competent C57BL/6 FUCCIgreen cell cycle reporter mice were isolated and stimulated in B cell medium for 48 h with LPS (25 µg/mL) + IL-2 (180 U/mL), LPS (12.5 µg/mL) + IL-2 (180 U/mL) + PMA (5 ng/mL) or left untreated. Green fluorescence was measured using a 470 nm excitation filter and a 510 nm emission filter. Between 1 × 10^5^ and 2 × 10^6^ FUCCIgreen^+^ and FUCCIgreen-negative CD19^+^ B cells per mouse were sorted by FACS (anti-mouse CD19 PE-Cy7 (eBio1D3)), and RNase H2 activity was measured in triplicates.

### 2.5. Flow Cytometric Cell Sorting

PBMC from human donors were stained with anti-human CD3 (UCHT1) PE and anti-human CD19 (SJ25-C1) APC-H7, murine spleen cells with anti-CD19 (eBio1D3) PE, anti-CD4 (RM4-5) APC, anti-CD11b (M1/70) eF450 and anti-CD11c (N418) PE/Cy7, murine peritoneal lavage cells with anti-CD11b (M1/70) eF450 and anti-F4/80 (BM8) PE, and murine epidermal cells with anti-CD49f (eBioGoH3 rat) PE antibodies at 4 °C for 30 min. Antibodies were purchased from Thermo Fisher Scientific Germany (Frankfurt a. M., Germany). Stained cells, or harvested cell culture cells, respectively, were washed with FACS buffer (Table 1) three times and resuspended in FACS buffer. Shortly before analysis, DAPI was added to a final concentration of 3 µM. Cells were sorted on a BD FACSAria™ III (Beckton Dickinson Germany, Heidelberg, Germany) excluding doublets and dead cells. Data were analyzed using FlowJo Single Cell Analysis Software (FLOWJO, LLC Data analysis software).

### 2.6. Cell Lysis and Protein Quantification

Washed cell pellets were dissolved in a suitable amount of lysis buffer 1 (Table 1) and incubated on ice for 10 min. After addition of the same amount of lysis buffer 2 (Table 1) and another incubation on ice for 10 min, cell debris was spun down at 20,000× *g* for 10 min at 4 °C. Supernatant containing total cellular protein was harvested, and replicates were stored at −80 °C. Protein concentration was determined in duplicates using the Qubit™ Protein Assay Kit (ThermoFisher scientific) following recommendations of the vendor. Lysis buffers were based on Rigby et al. [49].

### 2.7. RNase H2 Activity Assay and Standard Conditions

RNase H2 activity was measured using a fluorometric assay approach adapted from Crow et al. [44]. The type 2 RNase H substrate consisted of an 18 bp DNA strand containing a single ribonucleotide 4 bp 5′ of a covalently attached 3′ fluorescein residue (oligonucleotide B) which was annealed to an 18 bp anti-sense DNA strand with a quenching 5′ dabcyl residue (oligonucleotide D, the double-stranded substrate is referred to as BD) (Table 2). Type 2 RNase H hydrolyzes the phosphodiester bond 5′ of the single ribonucleotide leading to dissociation of the fluorescein-carrying fragment from the quencher, which allows for photometric quantification (Figure 1A). Unspecific DNase-mediated substrate cleavage was controlled for by using a double-stranded DNA substrate with the same sequence but lacking the single ribonucleotide in order to make it type 2 RNase H-resistant (ED) (Table 2, Figure 1A). As positive controls, unquenched single-stranded substrates (oligonucleotides B and E), unquenched double-stranded substrate lacking the dabcyl residue at the anti-sense strand (BK), and plateau-fluorescence of the fluorescence progress curves (BD and ED plateau) were implemented (Figure 1A,B). As negative controls, quenched double-stranded substrates (BD and ED) without addition of enzyme, quenched type 2 RNase H-specific substrate with addition of *heat-inactivated* cell lysate (BD + h.i. lysate), quenched double-stranded 2-O′-methylated RNA:DNA (type 2 RNase H-resistant) substrate (AD) with addition of cell lysate, and blanks were used (Figure 1B). Desalted oligonucleotides were purchased from Sigma-Aldrich (Merck KGaA, Darmstadt, Germany), dissolved in TE buffer to a final concentration of 100 µM, annealed by heating to 90 °C for 2 min and then gradually cooled down by 1 °C per minute. Annealed substrates were aliquoted and stored at −20 °C at a concentration of 10 pmol/µL. 

After cell isolation, cell number and protein content were quantified, and cell pellets were lysed as described above. Cell lysates were premixed on ice with reaction buffer in a 96-well PCR strip plate. Then, equal amounts of cell lysate premixes were pipetted to a sterile, vacuum-sealed, RNase-free, transparent flat-bottomed 96-well reaction plate (Sarstedt AG & Co. KG, Nuernbrecht, Germany) containing pre-warmed reaction buffer (at 37 °C) with 27 pmol type 2 RNase H substrate (BD) or double-stranded DNA substrate (ED) using a multi-channel pipette. Total reaction volume was 115 µL, resulting in a total substrate concentration of 235 nM. The reaction was monitored in a FLUOstar^®^ Omega photometer at 37 °C for at least 120 min, and fluorescence was measured at 3 min intervals. 

Before measurement, the photometer was calibrated, setting the 30 pmol (260 nM) unquenched single-stranded substrate B positive-control fluorescence to 33,000 fluorescence units (FU). Photometer measurement range was set to 100%. A 485 nm excitation filter and a 520 nm emission filter were used. Before each measurement, wells were mixed by orbital shaking (3 mm diameter, 5 s). Fluorescence was induced by 10 flashes per well and cycle, and measurement was performed by orbital scanning with a vertically adjusted sensor. Fluorescence was measured using the fluorescence intensity approach. Positioning delay was set to 0.2 s and measurement start time to 0.5 s. All assay steps except for photometric measurement were carried out on ice. Calibrated fluorescence data was blank-controlled and converted into the equivalent amount of cleaved substrate BD (or ED), using the BD (or ED) plateau fluorescence standard curve (Figure 1C) to acquire substrate cleavage progress curves. The cleavage rate was obtained by linear regression of the curves between minutes 3 and 24 (at least 5 data points). Cleavage rates were corrected for spontaneous degradation of substrate BD and ED. Using an RNase HII and a Dnase standard with known activity (5 pipetting replicates of each standard in each experiment, see Figure 1H), substrate conversion standard curves (‘BD + Rnase HII’, ‘BD + Dnase’ and ‘ED + Dnase’, see Figure 1F,G) were calibrated for the experiment. The substrate conversion standard curve ‘ED + Dnase’ was used to match substrate ED cleavage rates of the cell lysates to the equivalent standardized catalytic activity of Dnase, relating 1 ‘eqU Dnase’ to the equivalent of the catalytic activity of 1 U Dnase (NEB^®^). The corresponding substrate BD cleavage rate through Dnase activity in the cell lysates was then calculated using the substrate conversion standard curve ‘BD + Dnase’. Finally, the proportion of substrate BD cleavage caused by Dnases was subtracted from total substrate BD cleavage rates of the cell lysates to obtain Rnase H2 activity of the cell lysates. This Rnase H2 activity was transformed into the equivalent standardized activity of Rnase HII using the substrate conversion standard curve ‘BD + Rnase HII’, referring 1 ‘eqU Rnase HII’ to the equivalent of the catalytic activity of 1 U Rnase HII (NEB^®^). For oligonucleotide sequences and buffer reagents, see Table 1 and Table 2.

### 2.8. Statistics

Statistical tests were performed using GraphPad Prism^TM^ 5.04 (GraphPad Software Inc., San Diego, CA, USA). Sample size calculations (two-sample *t*-test) were conducted as proposed by Hulley et al. [50] using G*Power Version 3.1.9.4 © (Franz Faul, Kiel, Germany).

**Figure 1 jcm-12-01598-f001:**
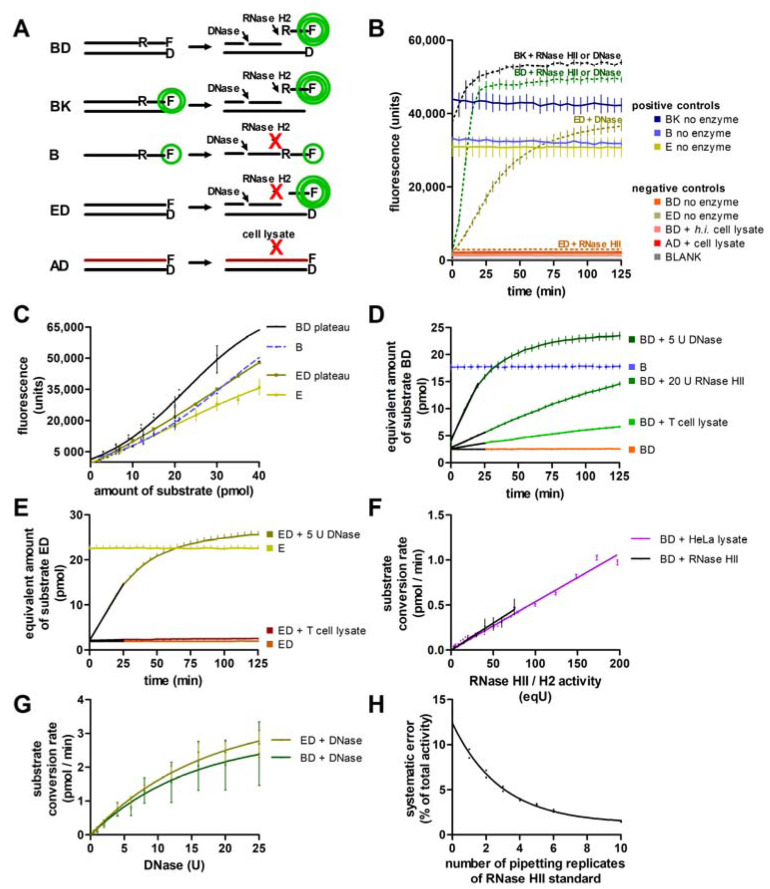
Assay principle and implementation of controls. (**A**) Substrate cleavage releases the fluorophore from the quencher. Details on substrates are shown in Materials and Methods. Substrates BD and BK are cleaved by RNase H2/HII and DNases. Substrates B and ED are cleaved by DNases only. Substrate AD is resistant to RNase H2/HII and was not cleaved by active cell lysate. Cleavage of substrate B did not alter fluorescence intensity (not shown); (**B**) addition of RNase HII or DNase (curves are shown for 5 U DNase) led to cleavage of positive control BK as well as substrate BD. DNase, unlike RNase HII, also cleaved substrate ED. Fluorescence of cleaved substrates BD, BK and ED showed sigmoidal increase until reaching a plateau after approximately 250 min (BD, BK or ED plateau, respectively). Fluorescence plateaus of substrates BD, BK and ED ranged above the fluorescence levels of positive controls B and BK. This aligns with a quenching effect of the complementary strand [51]. Positive control BK exceeded positive control B fluorescence when no enzyme was added. This is explained with decreased fluorophore stability of substrate B lacking the protective effect of the quenching complementary strand during storage [52]. Fluorescence negative controls included quenched substrates BD and ED without addition of enzyme, BD and ED with addition of heat-inactivated (h.i.) cell lysate, and quenched methylated substrate AD with cell lysate and blanks. Negative-control fluorescence reached a maximum of 6.1% (95% CI: 5.3%–6.9%) of BD plateau fluorescence. Unspecific substrate (BD) cleavage or degradation was insignificant with 0.1% per minute (95% CI: 0.0%/min–0.2%/min). Fluorophores showed stable fluorescence with a fluorescence decrease of 1.7% per hour (95% CI: 0.6%/h–2.8%/h). For graphical clarity of the negative-control fluorescence curves, these data are displayed with an offset; (**C**) implementation of fluorescence standard curves revealed concentration-dependent fluorescence non-linearity of the fluorophores. B, BD plateau, E and ED plateau fluorescence were measured in pipetting triplicates at eight different substrate concentrations (20–400 nM) after addition of 5 U DNase; (**D**,**E**) with help of the BD plateau fluorescence standard curve, BD (**D**) and ED **(E**) fluorescence progress curves were transformed into substrate cleavage progress curves now showing a perfectly linear segment indicating steady-state conditions of the pseudo-first order irreversible cleavage reaction. While addition of T cell lysate led to a clearly visible increase in substrate BD fluorescence, the same amount of T cell lysate barely resulted in substrate ED cleavage; (**F**) a substrate BD conversion standard curve was implemented using different amounts of E. coli RNase HII (NEB^®^, 4 U–75 U, shown in black). Using this curve and standards with known activity, measured catalytic activity can be assigned a standardized, externally validated unit (eqU). Doing so, the RNase HII standard curve was complemented by an RNase H2 standard curve using different amounts of HeLa cell lysate (8 eqU–200 eqU). The curves showed no significant deviation from linearity, indicating steady-state conditions; (**G**) substrate BD and ED conversion standard curves for DNase (NEB^®^, 0.5–25 U), in contrast, showed asymptotic rather than linear shape. Differences observed between the two curves were not statistically significant; (**H**) increasing numbers of pipetting replicates using equal amounts of RNase HII standard were used for calibration of the substrate BD standard curve. Systematic error between the experiments decreased with increasing number of matched standards; mean plus/minus SEM is shown.

## 3. Results

### 3.1. Implementation of Controls and Standard Curves

Assay principle and design of substrates were adapted from Crow et al. [44], and controls and assay procedure are described in detail in Materials and Methods (see also Figure 1). The assay was performed and validated in two independent laboratories, with similar results.

Fluorescence of all substrates was measured with and without the addition of enzyme (Figure 1B). Quenched substrates BD and ED reached 6.4% and 7.7% of unquenched positive controls B and E, respectively, indicating efficient quenching by the dabcyl group. Spontaneous dequenching by degradation of quenched substrate BD and ED was insignificant (Figure 1B). Addition of heat-inactivated cell lysate had no effect on fluorescence, indicating absence of unspecific (heat-sensitive) quenchers. No unspecific substrate degradation was detected upon addition of active cell lysate to 2-O′-methylated RNA:DNA substrate AD (Figure 1B). Blank fluorescence of the 96-well plate containing 115 µL of reaction buffer only did not differ by more than 500 FU between the different blank wells (1.6% of positive-control fluorescence). 

Addition of RNase HII or DNase increased fluorescence of substrate BD and BK, indicating substrate cleavage (Figure 1B, curves are shown for 5 U DNase only). Substrate ED was cleaved by DNase, but not by RNase HII. The observation that cleavage of substrate BK led to an increase in fluorescence despite lacking the quenching dabcyl residue can be explained by a quenching effect of the complementary strand [51]. Maximum fluorescence of fully cleaved substrate BD and ED (BD and ED plateau fluorescence) exceeded substrate B and E fluorescence, respectively. This is likely due to decreased fluorophore stability of substrate B lacking the protective effect of the quenching dabcyl residue and complementary strand during storage [52]. This would also explain why substrate BK showed higher fluorescence than substrate B. Plateau fluorescence of substrates BD and ED defined the standard curves for substrate conversion (Figure 1C). Substrate B positive-control fluorescence was used for photometer calibration.

Fluorescence progress curves were S-shaped, exhibiting a significant lag phase. This was unexpected for a pseudo-first order irreversible reaction with a single substrate and without any known inhibitors or any described conformational changes of the enzyme during the reaction [53,54]. Therefore, it was hypothesized that this lag phase was due to non-linear, concentration-dependent fluorescence behaviour of the fluorophore. Indeed, implementation of positive-control fluorescence standard curves using different amounts of substrate B and E and fully de-quenched substrate BD and ED (BD and ED plateau) demonstrated fluorescence non-linearity of the fluorophore (Figure 1C). We therefore converted fluorescence units into amount of cleaved substrate (pmol) based on the fluorescence standard curves (Figure 1C). Increase of amount of cleaved substrate over time showed perfectly linear behaviour without a significant lag phase (Figure 1D,E), allowing for definition of the steady-state phase, in which linear regression was performed to calculate RNase H2 activity (shown as black lines). 

To enable inter-laboratory reproducibility, the assay was validated using E. coli RNase HII and DNase with standardized activity. In three individual experiments, different amounts of RNase HII or DNase were added to samples containing substrate BD (RNase HII and DNase) or ED (only DNase) and fluorescence progress curves were determined in pipetting duplicates under standard assay conditions (Figure 1 F,G). Using these substrate conversion standard curves, measured catalytic activity was converted into the equivalent activity (eqU) of a defined amount of externally validated reference-RNase HII or DNase, respectively. We complemented the RNase HII substrate conversion standard curve using different amounts of HeLa cell lysate to obtain an RNase H2 substrate conversion standard curve under conditions of cell lysate assays (Figure 1F). Both standard curves showed no significant deviation from linearity, indicating steady-state conditions in the validated working range. In contrast, DNase substrate conversion standard curves featured an asymptotic curve shape, which might be due to enzyme saturation at higher substrate conversion rates than observed with RNase HII/H2 (Figure 1G). It was expected that DNase cleaves substrate ED more efficiently than substrate BD. However, differences observed between the two curves were not statistically significant. Inter-assay systematic error was assessed between three individual RNase H2 activity assays using seven aliquots of the same mouse embryonic fibroblast (MEF) lysate each. In each of the experiments, ten pipetting replicates using equal amounts of RNase HII standard with known activity were included for calibration of the substrate conversion standard curve (Figure 1F) to relate substrate BD cleavage to the corresponding standard catalytic activity (eqU). Systematic error of the calculated standard catalytic activities between the three experiments was dependent on the number of pipetting replicates used. With six pipetting replicates, systematic error decreased to less than 3% of total activity.

### 3.2. Sensitivity and Ruggedness

The calculation of the limit of detection (LOD) and the limit of quantification (LOQ) were performed according to Magnusson et al. [55]. Average curve slopes of four replicates of substrates BD and ED without addition of enzyme were 1.5 (SD = 1.3) and 1.2 (SD = 0.9) FU per minute, respectively (0.03% of positive-control fluorescence). This was equivalent to a substrate cleavage rate of 0.8 (BD, SD = 0.9), or, 1.6 (ED, SD = 1.5) fmol/min, respectively. The resulting LOD for substrate BD cleavage was 2.7 fmol/min (0.5 eqU RNase HII or 0.02 eqU DNase). The LOD for substrate ED cleavage was 4.5 fmol/min (0.02 eqU DNase). The LOQ for BD cleavage was 27 fmol/min (4.5 eqU RNase HII or 1.5 eqU DNase). ED cleavage had an LOQ of 45 fmol/min (2 eqU DNase). LOD and LOQ are summarized in Table 3.

Designing the RNase H2 assay for high-throughput analysis required prolonged sample handling and sample storage. Systematic sample handling error was assessed by subjecting HeLa whole cell lysate to repeated freezing and thawing (Figure 2A) or by incubation at RT for a defined time (Figure 2B). Each freeze-thaw cycle reduced RNase H2 activity by 3.1%. Likewise, incubation at RT reduced enzyme activity at a rate of 4.4% per hour.

Standard sample handling involved one freeze-thaw cycle and incubation times between 30 min and 2 h, which were, however, performed on ice rather than RT, suggesting maximal loss of RNaseH2 activity of 11.9% due to the processing.

### 3.3. Steady-State Kinetics and Assay Endpoints

In searching for the most-suited parameter to be determined (assay endpoint) for purposes of clinical screening, RNase H2 steady-state kinetics was studied. For this, 2.5 µg of HeLa protein (8 eqU RNase HII) from six individual HeLa cell cultures was added to wells containing eight different substrate concentrations spread around the expected K_M_-value [6,56]. Michaelis-Menten curves were determined using the initial-rates method and Michaelis-Menten non-linear regression (Figure 3). RNase H2 activity followed Michaelis-Menten kinetics with a mean K_M_ of 123 nM and a high mean C_V_ of 65.7%. Calculation of V_MAX_ was less variable with a mean C_V_ of 24.5%. Still, highest precision was obtained by measuring RNase H2 activity at a single substrate concentration (provided a substrate concentration > 2 SD above K_M_ was used). At a substrate concentration of 235 nM, RNase H2 activity reached approximately 60% of V_MAX_ and measurement variability was below 10%. Higher substrate concentrations lead to a systematic distortion of RNase H2 activity due to non-linear fluorescence behaviour of the fluorophores (Figure 1C). Therefore, we concluded, that RNase H2 activity at a substrate concentration of 235 nM was best suited as assay endpoint for purposes of clinical screening.

### 3.4. Precision

Overall precision was determined for the assay based on primary or cultured cells. Hereby, the influence of different assay steps (cell isolation and preparation, pipetting, photometric measurement and linear regression) and normalization methods (normalization to cell number or total protein) on overall variability was assessed. Total assay variability including all biological and methodological error sources ranged from 8.6% to 16% (C_V_).

Variability due to linear regression, photometer imprecision or pipetting was assessed in an experiment with 105 pipetting replicates and averaged at 7.7%. This accounted for more than half of total assay variability (56.2%; 95% CI = 14.7%–97.8%, *n* = 4), depending on the experimental approach and normalization method (contribution of single error sources to the total coefficient of variation was calculated using addition of variances). Under standard conditions, the largest part of this methodological error was attributed to pipetting error, while linear regression and photometer imprecision constituted minor error sources (not shown). Assay precision was strongly dependent on the normalization method used. Experiments relying on normalization to total protein averaged a total assay variability of 9.6%, unaffected by the cell isolation method (FACS-sorting or direct lysis of cultured cells), while normalization to cell number resulted in a much larger total assay variability of 16% (Figure 4). We attributed this primarily to loss of cells during washing steps. Isolation of primary cells from peripheral blood by Ficoll^®^-gradient centrifugation is known to yield PBMCs of variable cell-type composition and viability [57,58]. Thus, isolation of primary blood cells resulted in higher overall variability (11.2%) than direct lysis of cultured cells.

### 3.5. Screening RNase H2 Activity in Human Lymphocytes

Total substrate BD cleavage activity differed significantly between cell types (Figure 5). In mouse cell lysates, BD cleavage was higher in S-/G_2_-/M-phase cycling cells (FUCCIgreen^+^) compared to cells in G_1_ and non-cycling cells (FUCCIgreen-negative) (Figure A1). To reduce variability, we aimed to assay for enzyme activity in one particular cell type rather than in samples containing undefined mixtures of different cells. We chose to base the assay on lymphocytes as they are easily obtained in large numbers from blood and feature high RNase H2 activity (Figure 5 and Table A1). PBMCs were obtained from blood samples of 24 healthy donors (one single cell isolation (Ficoll^®^-gradient) replicate and pipetting replicate each, Figure A2) with unknown *RNASEH2* genotypes by Ficoll^®^-gradient centrifugation. CD19^+^ B cells and CD3^+^ T cells were isolated by flow cytometric sorting. From a 10 mL blood sample, approximately 4.0 × 10^5^ B cells and 3.0 × 10^6^ T cells were obtained, sufficient for multiple replicate measurements (Table A1). To perform one assay with two cell isolation (Ficoll^®^-gradient) replicates, each analyzed in pipetting duplicates, 3 mL (B cell assay) or 1.5 mL (T cell assay) of venous blood was sufficient. Control group size was designed to allow detection of a reduction of RNase H2 activity by 30% with a statistical power of 90% and *α* of 0.10.

In T cells, substrate BD cleavage activity per µg of cellular protein was about 3-fold higher than in B cells (Figure 6A), while cleavage activity per cell did not differ significantly between B and T cells (Figure 6B), reflecting higher total protein content of B cells compared to T cells. *Inter*-individual assay variability in T cells was approximately four-fold lower as in B cells, irrespective of the normalization method (Figure 6C). 

When activity was measured in T cells with normalization to cell numbers, the highest error source was methodological variability (including cell isolation (Ficoll^®^-gradient) and pipetting errors), which accounted for 63% of total variance. *Intra*-individual variability (variation of substrate BD cleavage activity between cell lysates obtained from the same control group individual at five different time points in quadruplicates, corrected for methodological variability by addition of variances) contributed 26%, and inter-individual variability 11% to total variance. With normalization to total protein, intra- and inter-individual differences were much larger, explaining 80% of total variance. In B cells, inter-individual variability clearly exceeded intra-individual and methodological variability (Figure 6C). Gender and age did not contribute to inter-individual variability (Figure A2). 

In T cells, RNase H2 accounted for 92.5% (95% CI: 87.5%–97.5%) and DNases for 7.5% (95% CI: 2.5%–12.5%) of substrate BD cleavage (Figure 6D). Inter-individual variability of cellular Dnase activity did not exceed methodological variability and thus did not contribute to inter-individual variability of substrate BD cleavage activity.

Collectively, we show that quantification of Rnase H2 activity in T cells requires small amounts of venous blood and shows little *inter*-individual and *intra*-individual variation, making it a suitable method for clinical use. 

### 3.6. Reduced Rnase H2 Activity in T Cells of Patients with Systemic Autoimmunity 

A pilot experiment was performed on one systemic sclerosis patient (SSc1) and two systemic lupus erythematosus patients (SLE1 and SLE2). All patients were carrying mutations in one of the Rnase H2 genes known to impair enzyme activity or integrity in vitro (Figure 7A). SLE1 and SLE2 were the same individuals as described by Günther et al. [25]. SLE1 was a 50-year-old female heterozygous for the splice site mutation *RNASEH2C* c.348 + 1G>A (F116fs). The mutation was previously shown to impede Rnase H2 complex formation, resulting in reduced nuclear localization and complete loss of Rnase H2 activity in vitro [25]. The phenotype of this patient included rash, chilblains, photosensitivity, serositis, alopecia, recurrent fever, autoimmune thyroiditis and lymphopenia. At disease onset, she tested positive for ANA (anti-dsDNA) antibodies. At the time of this study, she received hydroxychloroquine and baricitinib. Patient SLE2 was heterozygous for the mutation K233Q (c.697A>C) in *RNASEH2B.* Experiments with recombinant enzyme showed a 75% reduction of activity compared to wildtype by this mutation [25]. She was a 40-year-old female presenting with the clinical features discoid cutaneous lupus, photosensitivity, arthritis and autoimmune thyroiditis. Serological findings included ANA (anti-Ro). She was treated with hydroxychloroquine and belimumab. Patient SSc1 carried a *RNASEH2C* variant with so far unknown effects on Rnase H2 activity (c.468G>T, heterozygous). The variant was identified by whole-exome sequencing and verified by Sanger sequencing of DNA from PBMCs (not shown). The condition of this 60-year-old female was classified as ‘limited disease’ (onset with 33 years) with antinuclear antibodies (titre 1:2560, Scl 70), Raynaud’s syndrome, mutilation of fingertips by digital ulcers at onset of the disease, lung fibrosis and oesophageal involvement. She received no immunomodulatory or immunosuppressive therapy. 

T cells of duplicate blood samples were collected at three (SLE2) or two (SLE1) different time points separated by at least two weeks. T cells of patient SSc1 were collected from three individual blood samples drawn on the same day. Blood sampling was timed between 7:00 and 10:00 a.m. for all patients and controls. Substrate BD cleavage activity was measured in cell isolation (Ficoll^®^-gradient) duplicates and compared to the group of healthy controls (*n* = 24, Figure A2) using unpaired one-tailed *t*-tests with Welch’s correction. RNase H2 accounted for 93% of T cell total BD cleavage activity. No significant differences were observed between patients SLE1 and SLE2 and five individuals of the control group (Figure 7B). DNase activity of all cell lysates was below the LOQ (0.004–0.019 eqU DNase). Substrate BD cleavage activity normalized to cell number was significantly reduced in T cells of all three patients compared to the control group (Figure 7C). Residual activity was 79% (SD = 5.9%) in SSc1, 40% (SD = 15%) in SLE1, and 65% (SD = 19%) in SLE2. When cleavage activity was assessed per µg of cellular protein, the effects were no longer significant for SLE1 and SLE2 (Figure 7D). Total amount of protein per cell did not differ significantly between patients and controls (Figure 7E). 

Collectively, our data show that the assay found reduction of RNase H2 activity caused by the heterozygous variants *RNASEH2B* K233Q, *RNASEH2C* c.348G > A, and *RNASEH2C* c.468G > T.

## 4. Discussions

We present standardization and validation of an assay allowing for quantification of RNase H2 activity in cell lysates. The assay is based on the experimental procedure published by Crow et al. [44], and is suitable for screening of clinical samples. To date, RNase H2 activity measurement has been performed primarily with recombinant wild-type or mutant RNase H2 [6,16,38,39,59]. Assays based on recombinant protein have important limitations since effects of, e.g., protein stability or protein-protein interaction are not captured. Recombinant protein expression requires sequencing and cloning of gene variants, precluding adaptation to settings of clinical screening. Assays involving overexpression of recombinant proteins also do not address effects of intracellular expression levels of functional enzyme. We directly measure RNase H2 activity of cell lysates, enabling determination of enzyme activity per cell or per amount of cellular protein, capturing any effect on levels of functional enzyme in the cell, including alteration of transcription, posttranscriptional regulation, posttranslational modifications and protein stability.

While the assay is based on lymphocytes sorted by FACS in the present study, less sophisticated methods of cell separation are clearly sufficient. In contrast to flow cytometric sorting, immunomagnetic cell separation [60] is cost- and labor-efficient, requires only simple equipment and allows fast enrichment of B or T cells from peripheral blood samples to high purity. 

We found that T cells are a suitable cell type for the detection of reduced RNase H2 activity in patient cells. Since RNase H2 is a nuclear protein acting on genomic DNA and cell volume is variable, activity per genome (per cell), seems to be a more important parameter than activity normalized to total protein. When RNase H2 activity was normalized to cell number, methodological variability (including Ficoll^®^-gradient and pipetting errors) was the largest error source in T cells. Combined methodological and *intra*-individual variability accounted for 89% of total control group variability. We concluded that, assuming only small variance differences, this would allow comparison of sample replicates of a single patient with the mean of the control group. In addition, we note that individual data points showing RNase H2 activity of the control group in Figure 7 cannot be equated with actual RNase H2 activity of individuals in the control group, as RNase H2 activity was measured in one Ficoll^®^-gradient and pipetting replicate only.

Measurement of RNase H2 activity in T cell samples collected at different time points from two patients with known mutations (SLE1 and SLE2) revealed significantly reduced activity when normalized to cell number [25]. Likewise, T cells of a patient suffering from systemic sclerosis showed reduced RNase H2 activity, independent of whether activity was normalized to cell number or total amount of cellular protein. However, statistical evaluations were difficult in this case since sampling replicates of patient SSc1 were all obtained at the same time point, and thus, intra-individual error was not controlled for in patient SSc1. Given the small number of control group individuals and replicates, the obtained effect sizes of a 21% (SSc1), 60% (SLE1) and 35% (SLE2) decrease of RNase H2 activity compared to the control group mean seem to be the most relevant parameter here. 

Future establishment of a larger and more heterogeneous control cohort evenly distributed between all age groups and gender is necessary to identify potential confounders (e.g., ethnicity, hormonal changes, medication, stress, circadian rhythm, CD4+/CD8+ ratio, etc.) contributing to inter- and intra-individual variability. Moreover, taking at least two samples of each control group individual at different time points would reduce total control group variability and facilitate determination of a cut-off below which RNase H2 deficiency is likely clinically relevant.

Due to the presented standardization and the implementation of externally validated standards meeting all requirements for certified reference materials, the assay ensures inter-laboratory reproducibility. High sensitivity, robustness against impact of sample storage or freezing, and a broad working range enable versatile applications.

Collectively, we provide a fully standardized, validated and benchmarked assay suitable for quantification of RNase H2 enzyme activity in clinical cell samples. The assay is sensitive and precise. It revealed differences in RNase H2 activity depending on cell type and cell cycle phase as well as the reduction of enzyme activity caused by a heterozygous *RNASEH2C* partial loss-of-function mutation. The assay will be valuable for screening clinical entities for alterations of RNase H2 function in autoimmunity and cancer.

## Figures and Tables

**Figure 2 jcm-12-01598-f002:**
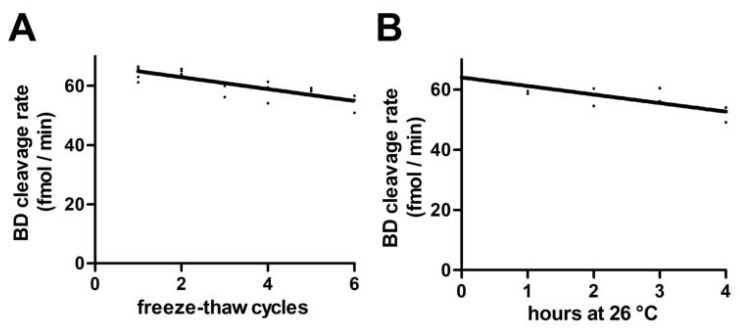
Ruggedness. (**A**) Aliquots of the same sample were frozen at −20 °C and thawed up to 6 times. Each cycle of freezing and thawing resulted in a mean activity loss of −3.1% (Pearson’s r = −0.94). (**B**) Aliquots of the same sample were stored at 26 °C for up to 4 h and activities were compared with the RNase H2 assay. Mean loss of activity was −4.4%/hour (Pearson’s r = −0.91).

**Figure 3 jcm-12-01598-f003:**
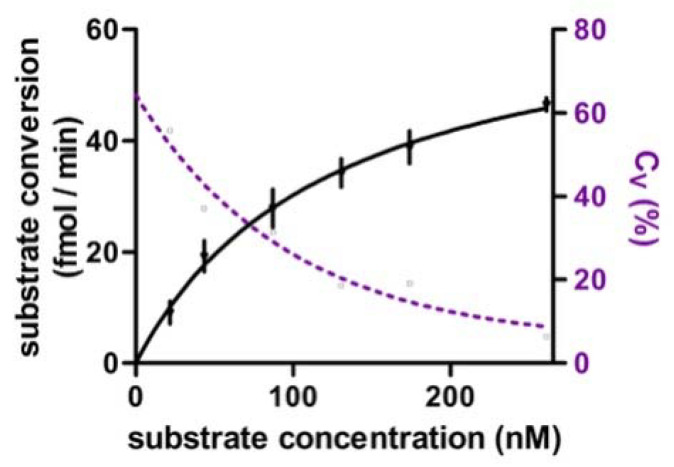
Steady-state kinetics. The mean K_M_ was 123 nM (SEM = 27.9%; 95% CI: 60–186 nM). Calculation of K_M_ and V_MAX_ values showed high variability (C_V_ (K_M_) = 65.7%; C_V_ (V_MAX_) = 24.5%), while the C_V_ of individual measurements was dependent on the substrate concentration. RNase H2 activity at a substrate concentration of 235 nM (>2 SD above K_M_) was associated with a C_V_ below 10% and implemented as assay endpoint; mean ± SD is shown.

**Figure 4 jcm-12-01598-f004:**
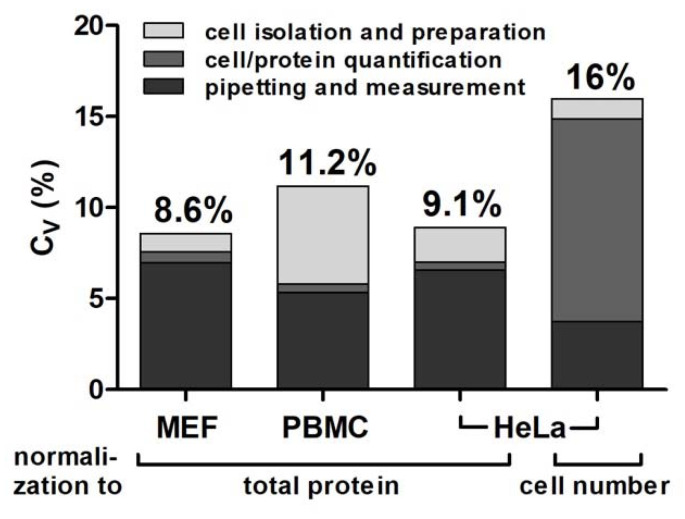
Assay precision. Substrate BD cleavage activity was determined in lysates of mouse embryonic fibroblasts (MEFs), lysates of PBMCs isolated from human blood by Ficoll^®^ gradient centrifugation, lysates of HeLa cells (sorted for living cells via FACS) with normalization to total protein or cell number as indicated. Total assay variability of the different approaches is shown above the bars. Stacked colors indicate contributions of different error sources (cell isolation and preparation; cell/protein quantification; pipetting and measurement imprecision determined in separate experiments (not shown)).

**Figure 5 jcm-12-01598-f005:**
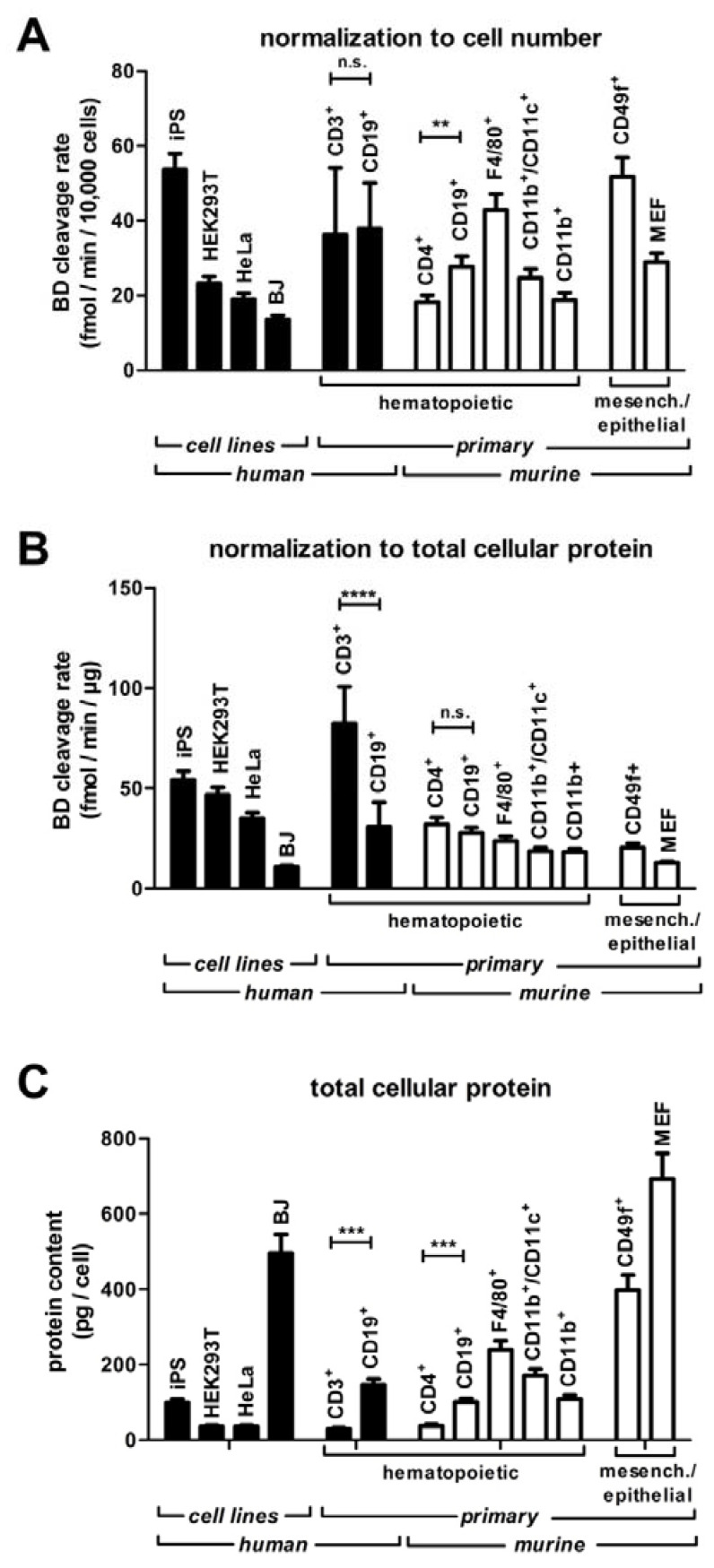
Total substrate BD cleavage activity of different cell types. Human-induced pluripotent stem (iPS) cells, human embryonic kidney 293 cells (HEK293T), HeLa cells, and human fibroblasts from the BJ cell line were cultured as described. Human peripheral blood T cells (CD3^+^) and B cells (CD19^+^), murine spleen T cells (CD4^+^), B cells (CD19^+^), dendritic cells (CD11b^+^/CD11c^+^) and macrophages (CD11b^+^), murine peritoneal macrophages (F4/80^+^), murine epidermal stem cells (CD49f^+^) and mouse embryonic fibroblasts (MEF) were purified as described above. Cells were counted by flow cytometry and then lysed. Protein concentration was determined and substrate BD cleavage activity was measured under standard assay conditions; (**A**) substrate BD cleavage rate normalized to cell number; (**B**) substrate BD cleavage rate normalized to amount of total protein; (**C**) total cellular protein content of the cell types; mean ± SD is shown for cell culture triplicates (cell lines iPS, HEK293T, HeLa, BJ), for the 24 individuals of the control group described below (human CD3^+^ and CD19^+^) and for pipetting triplicates of primary cell lysates from one mouse (CD4^+^, CD19^+^, CD11b^+^/CD11c^+^, CD11b^+^, F4/80^+^, CD49f^+^), or one mouse embryo (MEF); significance was tested with the unpaired two-tailed *t*-test: **** *p* < 0.0001, *** *p* < 0.001, ** *p* < 0.01; n.s. not significant.

**Figure 6 jcm-12-01598-f006:**
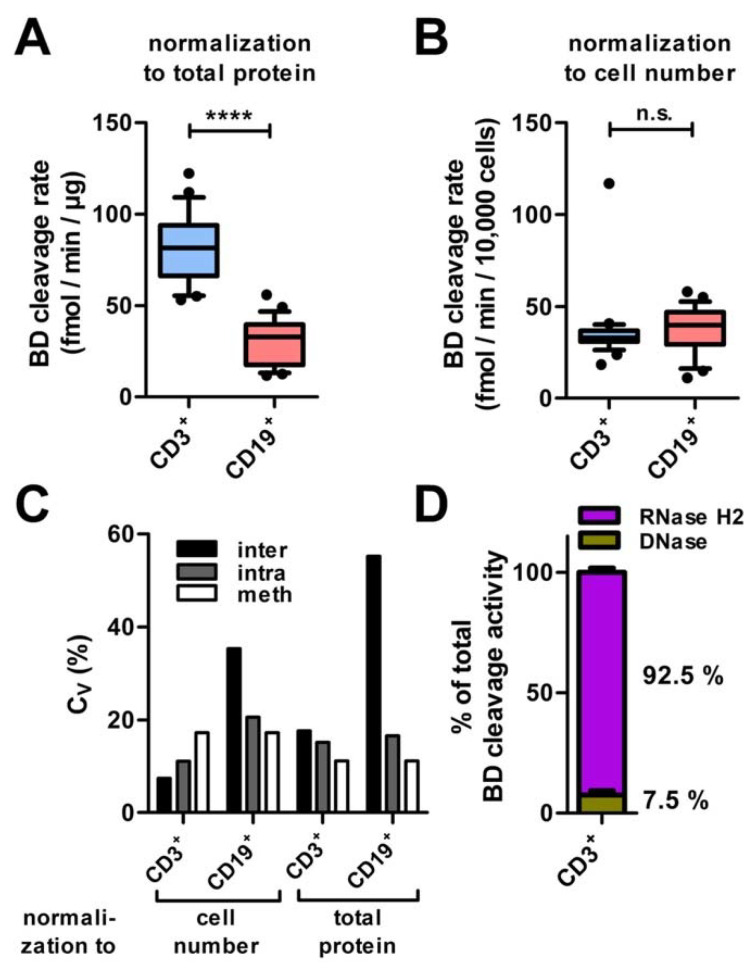
Control group benchmarks. (**A**) Substrate BD cleavage rates normalized to total protein in T cells were significantly higher than in B cells; activity was measured in single Ficoll^®^-gradient and pipetting replicates; (**B**) RNase H2 activity normalized to cell number did not differ significantly, but showed significantly smaller variability in T cells; (**C**) strong inter-individual variability was observed in B cells, while inter- and intra-individual variability were approximately on the same level as methodological error (including cell isolation (Ficoll^®^-gradient) and pipetting errors) in T cells. *Intra*-individual variability was assessed using quadruplicates of peripheral venous blood samples drawn from one healthy donor at five different time points. Methodological variability was determined in validation experiments; (**D**) the proportion of substrate BD cleavage activity due to DNase in the cell lysates was 7.5% of total cleavage activity (SD = 4%, *n* = 5); median is shown, boxes indicate 25th and 75th percentile, and whiskers indicate the 10th and 90th percentile. Normality could be assumed for all groups (Shapiro-Wilk test). Significance was tested via unpaired two-tailed *t*-test: **** *p* < 0.0001; n.s. not significant.

**Figure 7 jcm-12-01598-f007:**
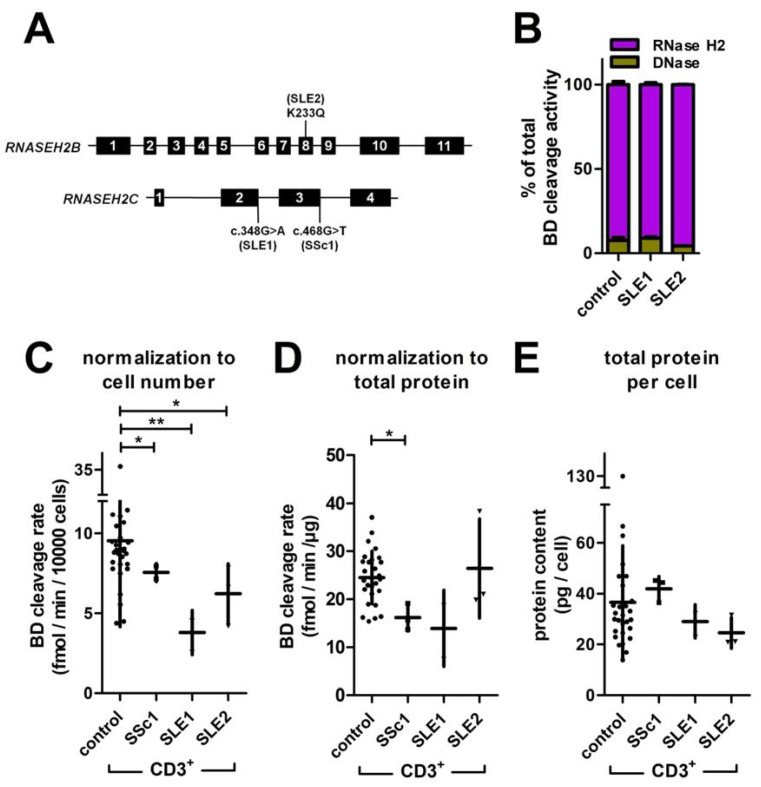
RNase H2 activity in T cells from SSc1, SLE1 and SLE2; (**A**) Mutation K233Q (SLE2) is located in exon 8 of *RNASEH2B,* while c.348G>A (SLE1) and c.468G>T (SSc1) are *RNASEH2C* splice site mutations at the end of exon 2 and 3, respectively, resulting in sterile transcripts [25]; (**B**) Rnase H2 activity accounted for 93% of total substrate BD cleavage activity in T cell lysates. Dnase activity was below the LOQ in all lysates. No differences were observed in patients and controls; mean ± SD is shown for five individuals of the control group (blood of each individual was sampled at two different time points in Ficoll^®^-gradient duplicates), two blood samples of SLE1 (Ficoll^®^-gradient duplicates) or three blood samples of SLE2 (Ficoll^®^-gradient duplicates); (**C**–**E**) total BD cleavage activity normalized to cell number (**C**) or to total cellular protein (**D**) and total cellular protein content (**E**) of T cells from SSc1, SLE1 and SLE2 compared to T cells from healthy controls (with unknown *RNASEH2* genotype, *n* = 24) Activity was measured using 3 µg of cellular protein (approximately 30′000–50′000 cells); mean ± SD is shown, and individual data points indicate single measurements of controls (a single Ficoll^®^-gradient and pipetting replicate each), three Ficoll^®^-gradient replicates of SSc1 (sampled at the same time), or the mean of duplicate Ficoll^®^-gradient replicates from each blood sampling (SLE1 and SLE2); significance was tested via unpaired two- tailed *t*-test with Welch’s correction: ** *p* < 0.01, * *p* < 0.5.

**Table 1 jcm-12-01598-t001:** Buffers.

Buffer	Reagents
reaction buffer (1×)	60 mM KCl, 50 mM Tris.HCl pH 8.0, 20 mM MgCl_2_, add fresh Triton X-100 and BSA to a final concentration of 0.01%
lysis buffer 1 (1×)	50 mM TRIS.HCL pH 8.0, 280 mM NaCl, 0.5% *v*/*v* NP40, 0.2 mM EDTA, 0.2 mM EGTA, 10% *v*/*v* glycerol, 0.1 mM sodium orthovanadate, add fresh 1 mM DTT and 1mM PMSF
lysis buffer 2 (1×)	20 mM HEPES, 10 mM KCl, 1 mM EDTA, 0.1 mM sodium or-thovanadate, add fresh 1 mM DTT and 1mM PMSF
FACS buffer	(1×) PBS, 3% FCS

**Table 2 jcm-12-01598-t002:** RNase H2 assay substrates. Oligonucleotides A, B and E align to the anti-sense oligonucleotides D or K.

Substrate	Sequence
oligonucleotide A: 2-O′-methylated RNA	5′-GAUCUGAGCCUGGGAGCU-fluorescein-3′
oligonucleotide B: DNA with a single ribouncleotide	5′-GATCTGAGCCTGGG[rA]GCT-fluorescein-3′
oligonucleotide D: DNA	5′-Dabcyl-AGCTCCCAGGCTCAGATC-3′
oligonucleotide E: DNA	5′-GATCTGAGCCTGGGAGCT-fluorescein-3′
oligonucleotide K: DNA	5′-AGCTCCCAGGCTCAGATC-3′

**Table 3 jcm-12-01598-t003:** Limit of detection (LOD) and limit of quantification (LOQ).

Substrate	LOD	LOQ
BD	2.7 fmol/min (0.5 eqU RNase HII or 0.02 eqU DNase)	27 fmol/min (4.5 eqU RNase HII or 1.5 eqU DNase)
ED	4.5 fmol/min (0.02 eqU DNase)	45 fmol/min (2 eqU DNase)

## Data Availability

Not applicable.

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
