# Peer review of "Development of an RNase H2 Activity Assay for Clinical Screening"

_jcm, 2023, doi:10.3390/jcm12041598_

Round 1
Reviewer 1 Report (Previous Reviewer 1)
The manuscript is significantly improved and all my previous comments have been addressed. The only recommendation of checking the description of Table 2, and figure legend of Figure A2 that contain some mistakes.
Author Response
Please see the attachment. Thank you very much for your supporting and helpful comments.

Reviewer 2 Report (Previous Reviewer 2)
The authors have significantly improved the manuscript, addressing most of my comments.
My main remaining concern is that I’m unsure whether this assay, as it stands, will be able to provide meaningful clinical information. This is mostly due to the imperfect data for controls used to compare RNase H2 activity in controls and patients (Fig 7C-E). The authors now explain much better what data they are working with and do discuss some of the limitations. I do appreciate the amount of work involved in generating a more ideal control dataset and don’t feel its absence should necessarily prevent publication at this stage.
However, because of these limitations I feel some of the statements should be further toned down. For example:
- Title: Better to not use “Validation”; perhaps “Development” would be more appropriate
- Abstract: “The assay readily detected reduced RNase H2 activity in lymphocytes of two patients with systemic lupus erythematosus and one with systemic sclerosis carrying heterozygous mutations in one of the RNASEH2 genes.” Perhaps re-phrase to something other than “readily detected”; same for the final sentence of Results
Minor:
- Line 349: “DNase cleaved substrate ED more efficiently than substrate BD”. This may be true, but the difference does not look to be significant to me. Please re-phrase.
- Fig 7B: Presumably SLE1 and 2 replicates are as described in the legend? What are the replicates for controls in this panel that give the mean and stdev?
- Line 789, “Combined methodological and intra-individual variability accounted for 89 % of total control group variability”. Given that only single samples were taken for the 24 controls, what intra-individual variability do the authors refer to?
- Line 811: “below which RNase H2 deficiency is clinically relevant”, probably better to use “below which RNase H2 deficiency is likely to be clinically relevant”
Author Response
Please see the attachment. Thank you very much for your supporting and helpful comments.

Reviewer 3 Report (New Reviewer)
This is a very detailed comprehensive paper for detection of RNase H2 activty associated with human diseases. 3 cases from patients are shwon as proof of concept . The standardiztion to cell number was statistically relevant but not dramtically strong. The usefulness of the assy for laboratories needs to be proven by applying the well-described technical procedure which is applicable by other laboratories.
The text is well-written and also the introduction to the field. I am not an expert on the clinical techniques but find them consistent and convincing. The number of cases will increase when other gourps follow this detection method and then it will prove its usefulness as a routine diagnostcis
accepted.
Author Response
Please see the attachment. Thank you very much for your review and comments.

This manuscript is a resubmission of an earlier submission. The following is a list of the peer review reports and author responses from that submission.
Round 1
Reviewer 1 Report
The manuscript from Schulz et al., described the validation for clinical screening of a previously described assay to determine RNase H2 activity in human patients.
RNase H2 deficiencies are associated with an increasing number of human disorders, from autoinflammatory and autoimmune disorders such as Aicardi-Goutières syndrome (AGS) and systemic lupus erythematosus (SLE) to cancer predisposition and outcome. Having a reliable and accurate RNase H2 assay would be a valuable tool for clinical diagnostic of diverse human diseases.
The assay they used is essentially the one described by Crow et al., (Crow et al., 2006) that consists of a fluorescein-labelled oligonucleotide containing a single ribonucleotide hybridized to a DNA oligonucleotide with a quencher. Upon cleavage, the fragment is released and free from the quencher and fluorescence is measure by photometry. Although the assay is not novel, the authors performed extensive validations, controls, and adjustments to detect RNase H2 in human and mouse cells. They concluded that for the clinical setting using human T-cells would be ideal, requiring small amounts of easily obtained blood and little variation in results. Then, they go to test their assay in one patient’s cells carrying a mutation in RNASEH2C, which is silent but affects splicing. They found a decrease in RNase H2 activity in the patient’s T cells, concluding that the assay identifies significant reduction of RNase H2 activity and could be used for clinical detection of RNase H2 deficiencies.
Major concerns:
· Because the RNASEH2C mutation they used to validate their assay has not been tested in vitro and no information about its activity is available, it’s hard to determine the robustness of the assay.
· I suggest using T cells from patients with RNase H2 mutations for which activity, expression and stability are known to see how the proposed assay correlates with previous data.
· To know more about the RNASEH2C mutation they tested, I would like to see western assays to see if protein expression correlates with activity.
Other concerns:
· The main assay is described in Materials and Methods (2.6), Results (3.2) and Figure legend 1, at length and with some discrepancies. In 2.6 they said in page 3 line 145-147 “BD plateau-fluorescence was determined by measuring fluorescence in wells containing 100 eqU HeLa RNase H2”. However, in 3.2 in page 6 lines 243-244: “Bacterial RNase HII cleaved substrate BD with fluorescence reaching a plateau (Figure 1B)”, and in legend of Figure 1, page 5 lines 210-212 they said, “Addition of 100U RNase HII lead to cleavage of positive control BK as well as substrate BD with fluorescence reaching a plateau (BK plateau and BD plateau)”. I suggest describing the entire process only once keeping simple and coherent.
· In page 6 line 261-263, they said: “implementation of a positive control fluorescence standard curve by enzymatic dequenching of substrates BD and B demonstrated fluorescence non-linearity of the fluorophore (Figure 1C)”. Which enzyme was used for substrate B and how dequenching works for this substrate?
· Figure 1B is very important for the manuscript, however it’s hard to evaluate. The blue colors of the BK and B substrates are very similar, the legend difficult to understand and error bars should be shown for the different curves.
Minor points:
- Some of the references in the introduction are not the most relevant in the field and even wrong. In line 40, they said: “While RNase H1 requires hybrids with at least 2 consecutive ribonucleotides”. In fact, it’s well documented that RNase H1 requires 4 consecutive ribonucleotides. It should be cited: DOI: 10.1016/j.cell.2005.04.024. In line 41 instead of reference [10], it should be used PMID: 1706718. In line 44: (Lujan et al., 2013; Sassa et al., 2019), these references don’t have the correct format. Also, Lujan et al., is not in the reference list. In page 1 line 48, instead of references (7-9) it should be used: DOI: 10.1016/j.molcel.2012.06.035. In page 1 line 86: “results in release of a fluorescein-labelled fragment from a quencher [41,42]”. Reference 41 used radiolabeled oligonucleotides no fluorescein labeled substrates. This reference should be removed from here and only use 42.
- Page 2, line 129 after lysis buffer 1, it should be (Table1), in line 130 after buffer 2 also say (Table1). Page 3 line 137 “42” should be (42). In line 139 after (oligonucleotide B) it should be (Table2).
- Page 9, supplementary Figure A2 is mention before Supplementary Figure A1. The figure first mentioned should be figure A1. Page 9, line 379, instead of Figure 6B, it should say Figure 6A. Line 380, instead of Figure 6C, it should say Figure 6B. Line 382, instead of Figure 6D, it should say Figure 6C.
Reviewer 2 Report
A FRET-based assay to measure RNase H2 activity has been widely used to measure nuclease activity of purified recombinant (mutant) RNase H2 protein, and has also been applied to measuring RNase H2 activity in cell lysates. The latter has been most commonly used to confirm loss of RNase H2 activity in KO cells, but has also been applied to confirm the impact of biallelic disease mutations in AGS patient cells (PMID: 23592335, 26903602, 35262626). Schultz and colleagues provide a comprehensive analysis of this assay applied to cell lysates of different human and mouse cells. They suggest ways of standardising the assay to allow comparison between different labs, with the intention of allowing its use as a validated clinical screening tool. This work provides a useful framework, providing others with information on how to perform this assay. Its application could provide clinically relevant information, for example to show the impact of (novel) RNase H2 mutations on cellular enzyme activity in patient cells. In case of biallelic mutations this may well be used to support a genetic diagnosis. However, in case of monoallelic variants this may not be as easy, and is perhaps more likely to indicate a greater or lesser likelihood of this variant being causative rather than incidental. An important question that remains is: at what point can it be concluded that RNase H2 activity below the mean of a number of healthy controls is clinically meaningful?
I have two major criticisms:
- An important control is missing from the assay. The normal substrate (BD) provides a measure of RNase H2 activity. The AD control provides some measure of non-specific activity, but a more appropriate control is double-stranded DNA (like BD, but no ribo), which controls for DNase activity against the BD substrate. As 2’-O-methylated RNA cannot be cleaved, only cleavage of the DNA strand or unwinding of AD will give background fluorescence. The important statement in line 248,249 cannot be checked as this data is not present in Fig 1B. Subtraction of background fluorescence with a dsDNA control will give a truer representation of RNase H2 activity in lysates (although still not perfect, as there will be some background due to non-specific cleavage at the ribo): it is much more similar to BD and will correct for cleavage of either strand as well as unwinding. This is important as some cells may have higher activity against dsDNA than others. Could this be part of the reason for the large variability seen for the B and T cell samples? It may not be feasible to repeat all assays with this control included and correcting RNase H2 activity for this background activity. However, at a minimum it should be discussed as a further improvement, and ideally experiments should be performed to show the impact of this correction for at least a few of the cell types. As the focus in terms of clinical testing is on T cells, these would be ideal to include.
- The large inter-individual variability in RNase H2 activity between T cells from controls makes high confidence predictions as to the clinical significance of reduced RNase H2 activity in cells from a single patient difficult. Depending on how it is expressed, the difference in RNase H2 activity between the highest and lowest controls is 2.5 to 14-fold. More importantly, ~10% of healthy controls have activity that is lower than that for the patient heterozygous for the RNASEH2C variant. This would suggest that the conclusion that this variant is clinically relevant has a ~10% chance of being incorrect. The authors correctly identify the need for a larger and more heterogenous control cohort. The impact of the large variability in control samples on interpretability of clinical samples and how this could be dealt with should be discussed in greater detail.
Other comments:
Methods
- Information on cells other than HeLa or MEFs is missing, including the source of FUCCI cells
- Composition of FACS buffer is missing
- Line 146, at this point in the text, the definition of “100 eqU HeLa RNase H2” is unclear
- Line 158,159, what is the purpose of 1:1 mixing? This essentially just dilutes the lysate further in lysis buffer. If this mix is then pipetted into 100ul of reaction buffer with 270nM substrate then the final substrate concentration is no longer 270nM. More detail is needed here to explain what is done exactly, including what the final reaction volumes are.
- What type of 96-well plates were used?
- Line 179, how is systematic error corrected for exactly? Please give more detail.
- Table 2, oligonucleotide A sequence, some of it is black (CUGGG), some grey. Does this mean something, or should it all just be black?
Results/discussion
- Fig 1B, some colours are too similar and difficult to distinguish. Also, AD + lysate (mentioned on line 248,249) is not present
- Fig 1F legend (Line 2357-240), it is not clear to me what the mean systematic error refers to (n=3) and how this corresponds to the six matched quality controls, what these are exactly and how these are used
- Line 367, it cannot be concluded that “RNase H2 activity changed with cell cycle phase” based on the FUCCI experiments. FUCCI- vs FUCCI+ shows proliferating vs non-proliferating cells. FUCCI+ cells are in cycle and can be red (G1) or green (S/G2/M). This should be described better (also in the legend of Fig A2)
- Line 383, define “intra-individual variability”
- Line 386/387, it seems unlikely that the data is sufficiently powered to make this statement. Regardless, provide gender and age RNase H2 data as part of Fig A1 to allow readers to judge for themselves.
- Fig 5, Indicate what number and type of replicates were performed to calculate mean and SD. It would be preferable if individual data points could also be shown.
- Fig 7, define what the repeats for the patient are (different blood samples or simply technical replicates of the same sample). Regardless, it is not good practice to perform a statistical test comparing biological replicates for many controls with (technical?) replicates for a single patient. The statements about statistical significance in line 428 and 472 should therefore also be moderated. See also major comments.
Appendix:
- Table A1, amount of protein, presumably this is in micrograms?
- Fig A2, more precisely define FUCCI+ (green/red?) and FUCCI–. Indicate what number and type of replicates were performed to calculate mean and SD. It would be preferable if individual data points could also be shown.